# Solar cycle, seasonal, and asymmetric dependencies of thermospheric mass density disturbances due to magnetospheric forcing

Andres Calabia[1, 2, 3], and Shuanggen Jin[1, 2*]

[1]School of Remote Sensing and Geomatic Engineering, Nanjing University of Information Science and Technology, Nanjing, 210044, China.
[2] Shanghai Astronomical Observatory, Chinese Academy of Sciences, Shanghai, 200030, China.
[3] Colorado Center for Astrodynamics Research, University of Colorado Boulder, Boulder, 80309-0431, Colorado, USA.

*Correspondence to*: SG Jin (sgjin@nuist.edu.cn; sg.jin@yahoo.com)

**Abstract.** Short-term upper atmosphere variations due to magnetospheric forcing are very complex, and neither well understood nor capably modeled due to limited observations. In this paper, mass density variations from 10-year GRACE observations (2003-2013) are isolated through the parameterization of annual, Local Solar Time (LST), and solar cycle fluctuations using a Principal Component Analysis (PCA) technique. The resulting residual disturbances are investigated in terms of magnetospheric drivers. The magnitude of high-frequency ($\delta$ < 10 days) disturbances reveals unexpected dependencies on the solar cycle, seasonal, and an asymmetric behavior with smaller amplitudes in June at the South Polar Region (SPR). This seasonal modulation might be related to the Russell-McPherron (RM) effect. Meanwhile, we find a similar pattern, but less pronounced at the northern and equatorial regions. A possible cause of this latitudinal asymmetry might be the irregular shape of the Earth's magnetic field (with the north dip pole close to Earth's rotation axis, and the south dip pole far from that axis). After accounting for the solar cycle and seasonal dependencies by regression analysis to the magnitude of the high-frequency perturbations, the parameterization in terms of disturbance geomagnetic storm time index $Dst$ shows the best correlation, while geomagnetic variation $Am$ index and merging electric field $E_m$ are best predictors in terms of time delay. We test several mass density models, including JB2008, NRLMSISE-00, and TIEGCM, and find that they are unable to completely reproduce the seasonal and solar cycle trends found in this study, and with a clear overestimation of about 100% during low solar activity periods.

## 1 Introduction

The connection between solar drivers and Magnetosphere-Ionosphere-Thermosphere (MIT) phenomenon is very complex and dependent on many processes. One of the most important processes is the variable solar wind plasma combined with a favorable alignment of the IMF (Interplanetary Magnetic Field), which can produce auroral particle precipitation at high-latitudes and increment of thermospheric Joule heating through coupled MIT processes linked to the Dungley cycle [*Dungey*,

1961]. This phenomenon can suddenly govern the structure and dynamics of the thermosphere, creating changes in mass density distribution through thermal expansion/contraction and changes in the composition of neutral species (i.e., $O/N_2$ depletion [*Lei et al.,* 2010]). For instance, solar flares increase the X-ray and Extreme Ultra-Violet (EUV) irradiance and produce nearly immediate energy-absorption, ionization, and dissociation of molecules. The occurrence of solar flares usually correlates to the rotational variation of the Sun (about 27 days and sub-harmonics at about 9 days, 7 days, and 5

days), resulting from the secular appearances of bright regions associated with sunspots that persist across solar rotations. Different open and closed magnetic flux domains in the solar corona provide different speeds and densities of the solar wind, forming an outward spiral with fast-moving and slow-moving streams. Fast-moving solar wind tends to overtake slower streams, forming turbulent Co-rotating Interaction Regions (CIR). In addition, Coronal Mass Ejections (CME) release fast-moving bursts of plasma at the corona of the Sun, which travels at higher speeds than CIRs. CIRs and CMEs create

magnetospheric storms, which are mainly driven by the electric fields linked to the Dungley cycle. CMEs can produce stronger storms than CIRs and are usually initiated by a southward IMF, enhancing the night-side convection, and increasing the ring current. When CIRs and CMEs reach Earth, the rapid increase in Poynting flux and particle precipitation along the Earth's magnetic field lines, originating from solar-wind/magnetosphere coupling processes, lead to an enhancement in Joule heating and disturbances in thermospheric composition, temperature, density, and winds (e.g., *Knipp et al.* [2013], *Lühr et al.*

[2004], *Lathuillere and Menvielle* [2004], *Sutton et al.* [2005]). First effects of CMEs and CIRs appear in the auroral zone as an increase in thermospheric mass density, and shortly afterward the perturbation propagates towards the equator, followed by a global expansion lasting from several hours up to several days.

Thermospheric mass density distribution, particularly during storm-time, is of great importance for Precise Orbit Determination (POD) of Low Earth Orbit (LEO) satellites, and for the understanding of MIT coupling. Aerodynamic-drag

associated with neutral-density fluctuations resulting from upper atmospheric expansion/contraction in response to variable solar and geomagnetic activity, increases drag and decelerates Low Earth Orbits, dwindling the lifespan of space-assets, and making tracking difficult. In addition, the energy transfer from the solar wind to the MIT system is complex, not completely described by models and observations, and many studies focus their efforts on a better understanding of all involved physical processes. Currently, atmospheric drag from mass density at LEO is the largest uncertainty in orbit determination and

prediction, because short-term variations produced by episodic solar activity, for example, are still not well modeled (e.g., *Marcos et al.* [2010]). Currently, Mass Spectrometer and Incoherent Scatter radar (NRLMSISE-00) [*Picone et al.*, 2002], Jacchia (JB2008) [*Bowman et al.*, 2008], the Drag Temperature Model (DTM-2013) [*Bruinsma*, 2008], and Thermosphere-Ionosphere-Electrodynamics General Circulation Model (TIEGCM) [*Qian et al.*, 2014] are some of the most representative models of mass-density variations in the upper atmosphere (in this work we compare our results with NRLMSISE-00,

JB2008, and TIEGCM). While the mathematical formulation used to model the vertical profile of NRLMSISE-00 is the exponential Bates profile [*Bates*, 1959], the Jacchia series use the arctangent function to represent an asymptotic behavior for the upper thermosphere. On the other hand, the first-principles TIEGCM physical model solves three-dimensional fluid

equations for the mutual diffusion of $N_2$, $O_2$, and O, including a coupled ionosphere, where the reactions involve ion species and energy budget, as well as self-consistent generation of middle and low latitude electric fields by neutral winds.

During the last decade, considerable progress has been achieved on observing and modeling responses to geomagnetic storms in the thermosphere. *Villain* [1980] firstly derived mass densities from the CACTUS accelerometer (a French acronym meaning ultrasensitive three-axis capacitive accelerometric detector), and *Bruinsma and Biancale* [2003] from the Challenging Minisatellite Payload (CHAMP) mission. Then, *Liu et al.* [2005] showed two structured arc-shaped enhancements of ~ 2000 km diameter in the auroral regions using CHAMP mass density estimates. *Liu and Lühr* [2005] and

*Sutton et al.* [2005] investigated the severe geomagnetic storm of November 2003 from CHAMP and, shortly afterward, *Bruinsma et al.* [2006] included the Gravity Recovery and Climate Experiment (GRACE) estimates to investigate the same storm, showing density increases up to 800 % in a few hours. *Rentz and Lühr* [2008] studied the climatology of the cusp-related thermospheric mass density anomalies as derived from CHAMP over 4 years (2002-2005), and showed an increase in density anomaly proportional to the square of the merging electric field $E_m$. *Sutton et al.* [2009] studied the response to

variations produced by the July 2004 geomagnetic storm from CHAMP, showing a time response significantly shorter than those used by the empirical models. Based on the variations of the ratio of density estimates between ascending and descending orbits, *Müller et al.* [2009] showed a slightly better parameterization of mass density employing the *Am* index instead of the *Ap* index, although the differences from using both indices still remain unclear. *Lathuillère, et al.* [2008] also showed a better correlation with the magnetic *Am* index than with the *Ap* index from one year of CHAMP-derived densities

and revealed a similar behavior between the day and the night side variations. Moreover, *Guo et al.* [2010] and *Liu et al.* [2010, 2011] investigated a large number of great storms ($Dst \leq -100$ nT) and showed similar correlations on the day-side to those on the night-side.

Concerning seasonal and inter-hemispheric asymmetries, several studies have identified and investigated the possible dependence on magnetospheric forcing. For instance, *Lu et al.* [1994] found a significant difference in the cross-polar-cap potential drop between the two hemispheres (when $B_z$ is positive and $|B_y| > B_z$), with a potential drop in the southern

(summer) hemisphere over 50% larger than that in the northern (winter) hemisphere. *Fuller-Rowell et al.* [1996] explained that a latitudinal asymmetry of the global thermospheric mass density distribution would be explained in terms of the prevailing summer to winter meridional flow, and *Forbes et al.* [1996] included to this explanation the different solar-driven meridional contributions at the day and night sides. *Fuller-Rowell* [1998] proposed a new mechanism based on huge

turbulent eddy mixing due to the seasonal inter-hemisphere thermospheric circulation, partially mixing the thermospheric species, and restricting their diffusive separation. The authors suggested that during solstice periods, this "thermospheric spoon mechanism" could be triggered by strong inter-hemispheric prevailing meridional winds originated by a global pressure gradient due to the asymmetric heating of the globe. The resulting inter-hemispheric asymmetry distribution of mass density could not be created during equinox periods, because of the resulting weak latitude pressure gradients and light

meridional winds from an equilibrium between the high-latitude and low-latitude sources of heating. More recently,

*Bruinsma et al.* [2006] also showed a latitudinal asymmetry with higher mass density values at the southern latitudes and suggested an enhanced summer versus winter Joule heating at high latitudes. *Ercha et al.* [2012] studied the hemispheric asymmetry of the thermospheric response to geomagnetic storms through a statistical analysis of 102 geomagnetic storms (2001 - 2007), showing much larger density enhancements in the South Polar Region (SPR) than in the North Polar Region (NPR). The authors attributed this phenomenon to the non-symmetric Earth's magnetic field. Moreover, *Deng et al.* [2014] examined the high latitude asymmetry in the Pedersen conductance from electron density profiles during 2008 to 2011, showing larger changes of energy partition between the ionospheric E (100-150 km) and F (150-600 km) regions in the southern hemisphere than in the northern hemisphere.

The above review describes the big picture of the current efforts done for a better understanding of mass density variations driven by magnetospheric forcing, and its modeling through correlations to their representative proxies. However, the sufficiently accurate set of drivers, proxies, and interrelations between involved geophysical processes is still incomplete, and more studies and modeling are needed. For instance, a proper removal of annual, LST, and solar cycle variation is key to unambiguously resolve the relation between proxies of magnetospheric forcing and mass density disturbances, and none of the previous authors have investigated a sufficiently large and continuous time-series of observations, at least to complete a solar cycle, while their statistical analyses were focused only on collections of large storms.

In this manuscript, we present a comprehensive study on thermospheric density disturbances due to magnetospheric forcing from 10-year (2003-2013) continuous time-series of GRACE accelerometer-based and POD-based mass density estimates, which have been isolated from annual, LST, and solar cycle variations through the parameterization of the Principal Component Analysis (PCA) [*Calabia and Jin*, 2016]. In this scheme, a continuous time-series can provide a more realistic representation during both active and quiet magnetospheric conditions, instead of analyzing a collection of large storms. The structure of this manuscript is presented as follows: Section 2 describes the data sets and analysis methods employed; Section 3 presents the results on dependencies and asymmetries from correlations and parameterizations at high and low latitudes; comparison with the current models as well as discussion are given in Section 4; and finally summary and conclusions are given in Section 5.

## 2. Data and analysis methods

### 2.1. Mass density estimates and geomagnetic indices

We employ thermospheric mass densities inferred from accelerometer and POD measurements made by the GRACE mission [*Tapley et al.*, 2004]. The GRACE satellites were launched into a nearly circular orbit on 17 March 2002 with an initial altitude of about 525 km, and a mean altitude of 475 km. The highly sensitive accelerometers onboard the GRACE satellites were initially designed to help to derive the Earth's gravity field, but the measurements of non-gravitational forces have

provided the unprecedented opportunity to derive and study thermospheric mass density variations. In this scheme, mass density estimates are obtained after removing irradiative accelerations from measured non-gravitational accelerations, where the resulting force is the combined effect of atmospheric drag and wind (aerodynamic) and can be expressed as a dynamic pressure applied on a reference area [*Jin et al.,* 2018]. The accuracy of the density observations under geomagnetic storm conditions is estimated to be about 10 - 40%, but since density perturbations are several hundred percents higher than that during quiet conditions, the uncertainty is still is acceptable for the purpose of this research. A more detailed description of the error budget is given by *Bruinsma et al.* [2004]. After mass density estimates are retrieved along orbits, the normalization to common altitude is performed with the use of an empirical model [*Bruinsma et al.,* 2006]. In LEO, the errors caused by the normalization of changes in altitude of ~100 km are expected to be within 5%, as discussed in *Bruinsma et al.* [2006]. In this study, we employ accelerometer-based and POD-based mass density estimates computed in *Calabia* [2016], and we provide the complete set at 3 min interval in the supporting information files [*Calabia and Jin,* 2019].

Space weather and geomagnetic indices are commonly used in upper atmosphere modeling during geomagnetic storms. These indices have been downloaded from the Low-Resolution OMNI (LRO) data set of NASA (http://omniweb.gsfc.nasa.gov/form/dx1.html) and from the International Service of Geomagnetic Indices (ISGI) website (http://isgi.unistra.fr/data_download.php). *Liu et al.* [2010] demonstrated that the merging electric field, $E_m$, is a physical quantity that closely correlates with mass density variations during geomagnetic storms. The merging electric field, $E_m$, assumes that there is an equal magnitude of the electric field in the solar wind, the magnetosheath, and on the magnetospheric sides of the magnetopause [*Kan and Lee,* 1979]:

$$E_m = v_{SW} \sqrt{B_y^2 + B_z^2} \sin^2\left(\frac{\theta}{2}\right) \tag{1}$$

where $B_y$ and $B_z$ are the IMF components, $v_{SW}$ is the solar wind speed, and $\theta$ the IMF clock angle in Geocentric Solar Magnetospheric (GSM) coordinates.

## 2.2. Parameterization of solar cycle, annual, and LST variations

In order to remove the solar cycle, annual, and LST variations from the initial density estimates, we have parameterized the main PCA modes of variability as done in *Calabia and Jin* [2016]. Other techniques such as wavelets are also widely employed, but in this work we employ a new two-step method ("PCA fit" + "fit of residuals") for more robust modeling. In brief, the purpose/aim of a PCA technique is to determine a new set of bases that capture the largest variance in the data, based on Eigen Value Decomposition of the sample covariance estimated from the initial data. Detailed analyses and the selection of retained modes for static grids can be found in *Preisendorfer* [1988] and *Wilks* [1995], and a readily computable algorithm in *Bjornsson and Venegas* [1997]. In this work, more data and a revised analysis have been performed with respect to the data provided by *Calabia and Jin* [2016]. For instance, we have included POD-based estimates to fill the data gaps of

accelerometer measurements [*Calabia and Jin*, 2017], and manually excluded obvious outliers caused by, e.g., geomagnetic storms and artifacts in the data processing. These improvements have provided a better representation of the variability with respect to *Calabia and Jin* [2016] results. The combined data set and model is provided in the supporting information files [*Calabia and Jin*, 2019]. The four leading modes together account for 99.8 % of the total variance and, individually, explain 92 %, 3.5 %, 3 %, and 1.3 % of the total variability. These high values indicate marked patterns of variability. The correlation coefficients between the parameterized time series of PCA modes and the initials are 96 %, 93 %, 90 %, and 83 %, respectively. These high values indicate high accuracy in the model. Then, in order to reflect the magnetospheric contribution through relevant proxies in the residuals, we employ a constant value of *Am*=6 in the parameterization given by *Calabia and Jin* [2016]. Herein refer the parameterization set of the solar cycle, annual, and LST variations as "radiation model" ($\rho_{model}$), while the residual disturbances ($\rho_r$) to mass density estimates ($\rho_{GRACE}$) are defined as:

$$\rho_r = \rho_{GRACE} - \rho_{model}$$

(2)

Assuming high efficiency in the model ($\rho_{model}$), the residual disturbances ($\rho_r$) will not only contain variations due to magnetospheric forcing but also disturbances due to other sources, as for example, lower atmospheric waves and recurrent Travelling Atmospheric Disturbances (TADs) [*Bruinsma and Forbes,* 2010]. These other disturbances are not regarded in this manuscript and can be investigated in future research after removal of the presented model. A more complete listing of known thermospheric mass density disturbances is given in *Liu et al.* [2017].

### 2.3. Density responses to magnetospheric forcing at different latitudes

Density residuals at poles and equatorial regions are extracted from the residual disturbances to determine their time-dependent relationship to changes in magnetospheric drivers. We denote $\rho_r$ in Eq. 2 as $\rho_E$ for the profile at the Equator, and $\rho_N$ and $\rho_S$ for the NPR and SPR profiles, respectively. Density profiles for each region (see example in Fig. 1a) correspond to an average value of density of a longitudinal band of 30° width in latitude, centered at Equator ($\rho_E$) and at the geographic poles ($\rho_N$, $\rho_S$). The Equator profile of density is computed as the mean average between ascending and descending orbits, so possible LST and time-lag differences between ascending and descending orbits are mitigated (although some studies introduced in Section 1.3.3. have shown a negligible LST contribution).

The approach employed in this work is based on the parameterization of the standard deviation, which provides a more robust metric modeling, instead of attempting to fit the direct signal of residual disturbances. Additional smoothing filters are applied to both residual disturbances and proxies as follows. First, we remove the disturbances longer than 10 days period ($\delta$) from both $\rho_r$ and geomagnetic indices to further standardize the data sets. We divide the approach in two steps, one for sub-daily variations, and the other for those between 1 and 10 days (we arbitrarily decided to employ the 10 days period since provided best results after several tests). The removal of longer trends is performed by subtracting the smoothed time-series

with a 10-day running-window filter, and the sub-daily variations are extracted in a similar way through a 1-day running-window filter. The general form for the mean running-window filter is:

$$Filt\left(x_i\right) = \sum_{j=i-a}^{i+a} \frac{x_j}{\left(2a+1\right)} \tag{3}$$

Where $x_i$ is the time-series to filter at each sampling index $i$, $a$ is half of the increment of time for each corresponding running-window, and $Filt(x_i)$ is the smoothed time-series employed to remove long-term signals.

The standard deviation shown in Fig. 2 is calculated for each pair of time-series ($\delta < 1$ day, and 1 day $< \delta < 10$ days) trough a 30-day running window (we employ a similar form of Eq. 3 to compute the standard deviation instead of the mean value). Fig. 2 reveal strong dependencies on the solar cycle, we further parameterize this dependence in terms of solar-flux F10.7

(Fig. 3, Table 1), and the results are given in Fig. 4. A seasonal dependency is shown with weaker disturbances during June solstice periods, mostly at the SPR. We parameterize this seasonal variation to better fit geomagnetic indices into the residual disturbances ($\rho_r$), by using the annual period in a Fourier fitting to the normalized disturbance in the SPR (Table 2). After this seasonal variation is removed, the normalized standard deviations of the residual disturbances ($\rho_r$) show good agreement with the standard deviations of the geomagnetic indices (see Fig. 4). The fitting of least absolute residuals (which minimizes

the absolute difference of the residuals) in a 2-variable parameterization (solar cycle and geomagnetic index) is chosen to better characterize singular events of strong geomagnetic activity:

$$\rho_r' = p_{00} + p_{10}\cdot Ind + p_{01}\cdot F + p_{20}\cdot Ind^2 + p_{11}\cdot Ind\cdot F + p_{02}\cdot F^2 +$$

$$+ p_{30}\cdot Ind^3 + p_{21}\cdot Ind^2\cdot F + p_{12}\cdot Ind\cdot F^2 \tag{4}$$

In this equation, $Ind$ correspond to the geomagnetic index employed ($A_m$, $Dst$, or $E_m$), and $F$ is the solar radio flux at 10.7 cm. As for the SPR, we easily modulate the parameterized northern profile in terms of the parameterized disturbances at the SPR, $\sigma''$ (Table 2):

$$\rho_S' = p_{00} + p_{10}\cdot \rho_N' + p_{01}\cdot \sigma'' + p_{20}\cdot \rho_N'^2 + p_{11}\cdot \rho_N'\cdot \sigma'' +$$

$$+ p_{02}\cdot \sigma''^2 + p_{30}\cdot \rho_N'^3 + p_{21}\cdot \rho_N'^2\cdot \sigma'' + p_{12}\cdot \rho_N'\sigma''^2 \tag{5}$$

Finally, Pearson's linear correlation coefficients between each profile of density disturbances and the final parameterizations are calculated with delay-times ranging from $\pm 18$ h. Analysis results are provided in the next section.

## 3. Results and analysis

This section is presented in three subsections. Firstly, an analysis of a single event is presented to exercise the typical storm-time behavior and to understand how proxies are employed to represent mass density variations. The second subsection represents the main contribution of this work, with the complete analysis of the 10 years time-series. Finally, an estimate of uncertainty and contrasts of results are made in the last subsections.

### 3.1. Analysis of a single event

Fig. 1a shows the northern ($\rho_N$), southern ($\rho_S$), and equatorial ($\rho_E$) profiles of mass density disturbances, normalized to 475 km altitude for the Moderate Geomagnetic Storm (rated as G2 in the NOAA's geomagnetic storm scale) of 18 March 2013. Shown are traces of mass density estimates with the solar cycle, annual, and LST dependencies removed. The following panels display the K-derived planetary indices ($ap$, $an$, $as$), the auroral index horizontal component disturbances ($AE$, $AL$), the solar wind ($SW$) velocity, proton and temperature, the longitudinally asymmetric horizontal component disturbances ($ASY\text{-}D$, $ASY\text{-}H$), the Electric Field ($Ey$), and the Polar Cap index horizontal component disturbances.

In general, density perturbations and space weather and geomagnetic indices remain calm until early morning ($\sim 5$ h UT) on 17 March 2013, and the geomagnetic storm commences. High latitude mass density profiles exhibit two peaks, a relative maximum at 10 h UT with $6 \cdot 10^{-13}$ kg/m$^3$, and an absolute maximum the same day at 17 h UT with $9 \cdot 10^{-13}$ kg/m$^3$. The equatorial variation shows a delay, starting the relative maximum at $\sim 7$ h UT, and reaching the absolute maximum of $4 \cdot 10^{-13}$ kg/m$^3$ at 22 h UT. The maximum of the equatorial density disturbance is less obvious but peaks at 10 h UT. In this figure, the $Dst$ index shows the best match with equatorial mass density disturbances. On the other hand, the best match for high latitude density disturbances is given with K-derived planetary indices ($ap$, $an$, $as$), $E_m$, and the auroral index horizontal component disturbances ($AE$, $AL$). This is expected based on locations of magnetometer stations that contribute to the corresponding indices. Finally, all indices start to return to the calm state at the end of the day ($\sim 24$ h UT), while the mass density profiles remain elevated until next day at $\sim 7$ h UT (18 March 2013). This phenomenon is the atmospheric response to equilibrate the global mass density back to the initial calm state. The question that arises from this figure is whether this is a typical storm-time behavior, and the extent to which this behavior could be modeled using their representative proxies in terms of time-delay, and other possible dependencies as, for example, an increasing in solar flux, or due to latitudinal asymmetries seen in previous studies (e.g., *Ercha et al.* [2012]; *Bruinsma et al.* [2006]; *Fuller-Rowell et al.* [1996]; *Forbes et al.* [1996]).

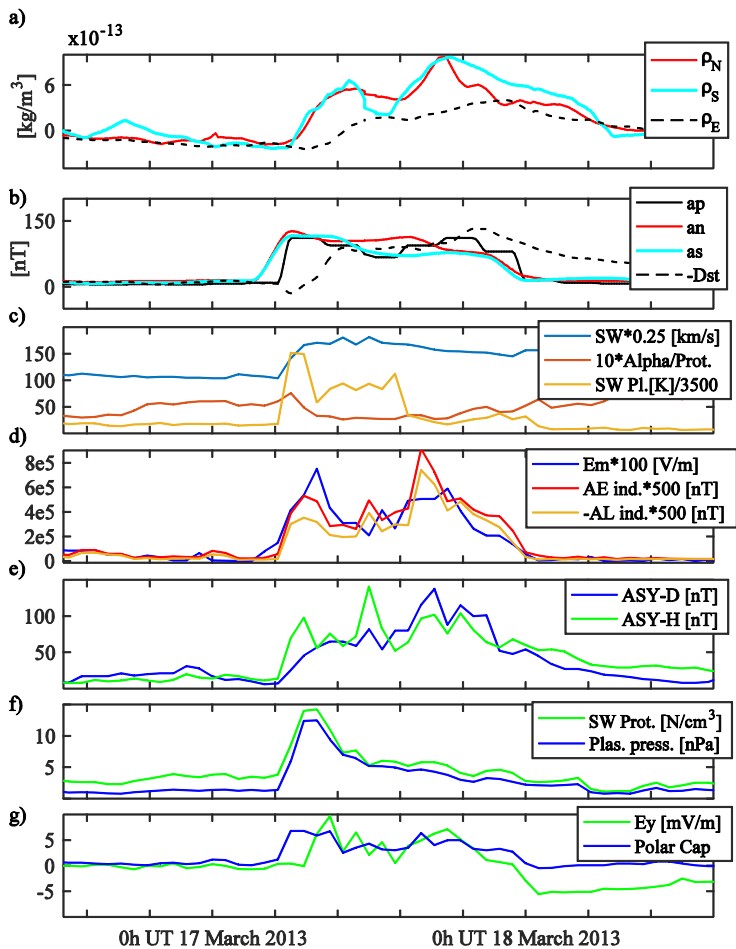

**Figure 1. In (a), the northern, southern, and equatorial profiles of residual density disturbances ($\rho_r$) are shown for the Moderate (G2) Geomagnetic Storm of 18 March 2013 (free from solar cycle, LST, and annual variations, and normalized to 475 km altitude). Space weather and geomagnetic indices are plotted below from (b) to (g). Magnitudes have been re-scaled as indicated in each legend.**

## 3.2. Analysis of 10-year thermospheric mass density time-series

In Fig. 2, the standard deviation calculated over a sliding window of 30-day length ($\sigma$) is shown for the NPR, Equator, and SPR ($\sigma_N$, $\sigma_E$, and $\sigma_S$) over the entire span of the 10-year analysis period. Possible unwanted variation with long-term periods, resulting from the process of removing solar cycle, annual, and LST variations have been eliminated by running a 10-day smoothing filter. In order to detect a possible variation of the geographical influence induced by the spatial component of the PCA model (e.g., location of magnetic dip, irregular magnetic field), standard deviations have been separately computed for the initial residual disturbances, and from both sub-daily and 1-10 days period ($\delta$) disturbances. As expected, sub-daily disturbances are smaller in magnitude when comparing to the longer periods. Concerning the latitudinal differences, disturbances at the southern region ($\rho_S$) are bigger in amplitude ($\sigma_S$) compared to the northern region ($\sigma_N$). The

values are described by the fitting in Fig. 3 and Tab. 1. At first look, the residual disturbances show strong alignment with the solar cycle trend (F10.7$_{81}$), indicating that the magnitude of mass density disturbances due to magnetospheric forcing is

255 strongly dependent on the 11-year solar cycle. Fig. 2 includes the 81-day averaged F10.7 solar-flux index to show the alignments. Note that these dependencies are intrinsic to the magnitude of disturbances due to magnetospheric forcing and not from the background LST, seasonal, and solar cycle variations which have been removed by the PCA model and smoothing procedures. In this scheme, the F10.7$_{81}$ index is firstly employed to fit the magnitude of disturbances, and the linear fits are presented in Fig. 3 and Tab. 1.

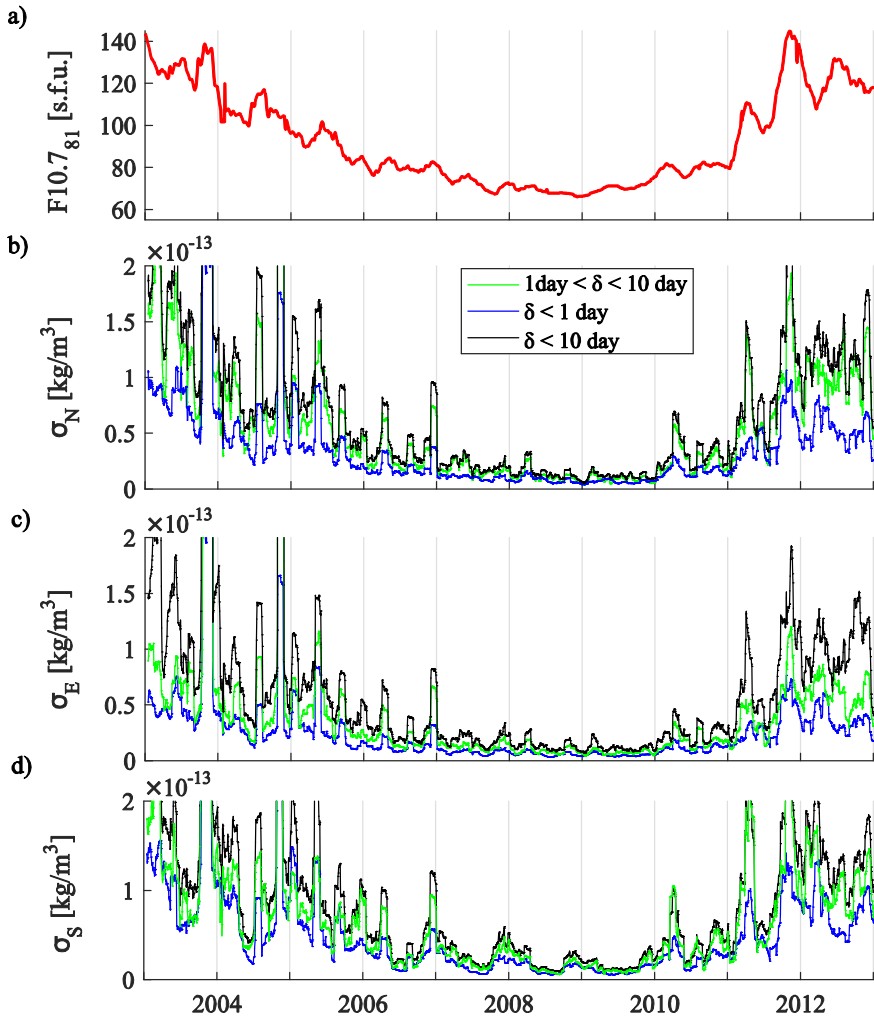

**Figure 2. From top to bottom, (a) the 81-day averaged F10.7 solar flux index, and the 30-day standard deviation sliding window of residual density disturbances at 475 km altitude in (b) the northern, (c) the equatorial, and (d) the southern regions. Calculations filtered at different frequencies are plotted in green, blue, and black.**

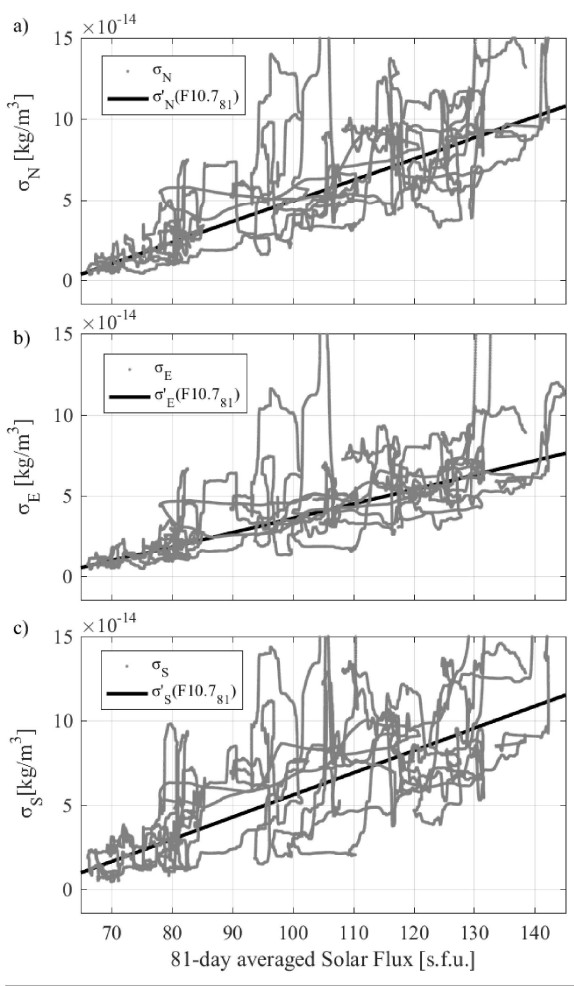

Figure 3. Linear fit of 30-day standard deviation sliding window of residual density disturbances at 475 km altitude from Fig. 2 at (a) NPR and (b) SPR with respect to 81-day averaged F10.7 solar flux index. Note that the seasonal variation (Table 3) has not been removed from (c).

**Table 1. Parameters and goodness of linear fit in Fig. 3. $\sigma$'(F10.7$_{81}$) = p1· F10.7$_{81}$ + p2.**

|  | $\sigma'_N$ | $\sigma'_E$ | $\sigma'_S$ |
|---|---|---|---|
| p1 | 1.3060E-15 | 8.8830E-16 | 1.3200E-15 |
| 95% conf. | (1.304e-15, 1.308e-15) | (8.871e-16, 8.895e-16) | (1.318e-15, 1.323e-15) |
| p2 | -8.1430E-14 | -5.2570E-14 | -7.6190E-14 |
| 95% conf. | (-8.161e-14, -8.125e-14) | (-5.269e-14, -5.246e-14) | (-7.645e-14, -7.593e-14) |
| R-square | 0.93 | 0.96 | 0.89 |
| RMSE [Kg/m$^3$] | 1.39E-14 | 8.99E-15 | 2.03E-14 |

An interesting feature in the resulting fit of Fig. 4 is the decrease of standard deviation in southern profile ($\sigma_S$) during June, clearly present in both sub-daily and 1-10 days time-series. On the other hand, the northern region $\sigma_N$ is more aligned with the F10.7$_{81}$ solar cycle variation, and without strong signs of a seasonal fluctuation (refer to the next section for more details). The discussion of the possible effects of this seasonal and asymmetric variation is given in the next section. The top panel in Fig. 4 plots the standard deviation computed with a 30-day sliding window of the *Am* and *Dst* geomagnetic indices, as well as the $E_m$. After accounting for solar cycle variation effects by data normalization using the parameters given in Tab. 1, the standard deviation computed using the same 30-day sliding window (three bottom panels in Fig. 4) shows a much better correspondence with the fitting of *Am*, *Dst*, and $E_m$ standard deviations. Though in the southern region (bottom panel in Fig. 4), lower mass density disturbances during summer seasons are now more obvious, and it has been parameterized in terms of day of the year (*doy*). The corresponding parameters and goodness of the Fourier fit are given in Tab. 2. We employ this parameterization to better characterize the fitting scheme of proxy candidates and additional dependencies. From Fig. 3 and 4, a clear contribution of the solar cycle for high and low latitudes is requisite for the fitting scheme (Eq. 4). In addition, the identified seasonal variation prominently modulates the fluctuations in the SPR. The last parameter to account for is the lag-time between proxies and density disturbances. Pearson's linear correlation coefficients between each profile of density and the final parameterizations are calculated with delay-times ranging from ± 18 h. Then, the maximum values for each time-series are employed in the fitting.

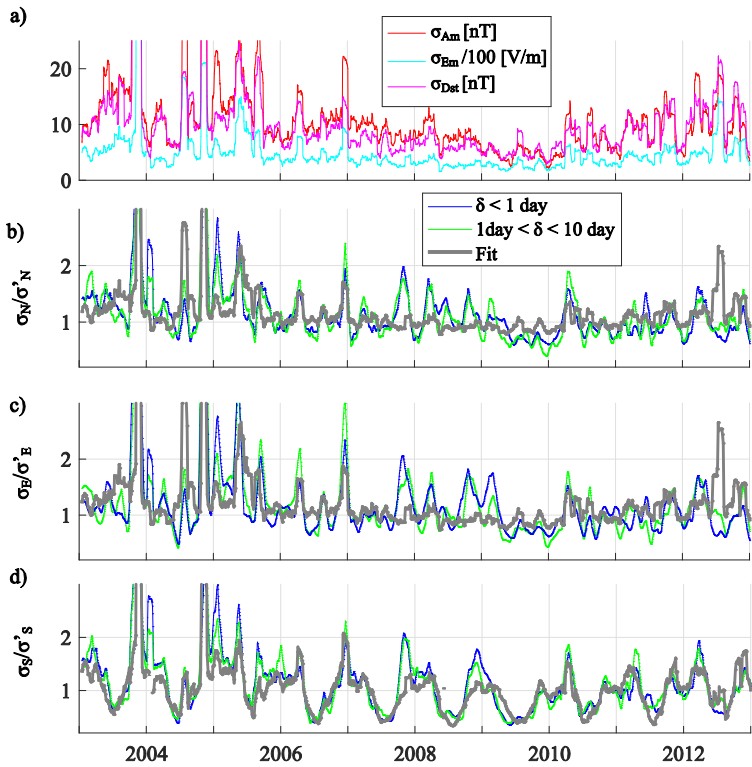

**Figure 4.** From top to bottom, (a) the 30-day moving standard deviation of *Am*, *Dst*, and $E_m$ , and the solar-flux normalized 30-day standard deviation sliding window of residual density disturbances at 475 km altitude at (b) the northern, (c) equatorial, and (d) southern regions. Fourier fits in terms of *doy* and $E_m$ are plotted in gray color. Calculations filtered at different frequencies are plotted in green, and blue.

**Table 2. Parameters and goodness of Fourier fit (Fig. 4). σ''(*doy*) = 1 + a1·**
**cos(*doy*) + b1· sin(*doy*) + a2· cos(2·*doy*) + b2· sin(2·*doy*)**

|  | $\sigma''_S$ |
| --- | --- |
| a1 | 0.3893  (0.3876, 0.3909) |
| b1 | 0.1043  (0.1027, 0.1059) |
| a2 | -0.1448  (-0.1464, -0.1431) |
| b2 | -0.05656  (-0.05817, -0.05494) |
| R-square | 0.80 |
| RMSE [kg/m$^3$] | 0.26 |

Fig. 5 plots the lag-time correlation coefficients between the parameterized perturbations and the three northern, equatorial, and southern profiles of residual disturbances ($\rho_r$). The top panel in Fig. 5 corresponds to residual disturbances on periods ranging from 1 to 10 days and the bottom panel corresponds to sub-daily disturbances. In general, sub-daily disturbances exhibit a smaller range of time-lag for correlations with drivers, than the longer variations shown in the top panel. A secondary maximum at about 12 h ahead of the absolute maximum might probably be originated by TADs reaching the opposite side of the globe, but further study is required to validate this assumption. Negative values proceeding to geomagnetic storms could also increase the secondary maximum. Negative values at equator prior to geomagnetic storms have been reported in earlier studies [*Calabia and Jin*, 2017], and further discussion in relation to the *Dst* index is given in the next section. Both time-delays of 1 to 10 days and sub-daily correlations show a similar response to the *Dst* index, showing a delay occurring after density disturbances, and revealing a shortcoming for prediction. On the other hand, *Am* and $E_m$ have much more capability as predictors. For the high latitude profiles, lag-time correlation of $E_m$ at sub-daily fluctuations has a double crest centered at the same time-lag as for the *Am* index. The lag-time correlation of *Dst* shows a potential capability of prediction for equatorial disturbances with periods shorter than one day. The most important feature in Fig. 5 is the correlation with *Am* and $E_m$ between 1 to 10 days, suggesting a great capability for prediction, as seen by the lag-time peak correlations of ~ 0.65 for high latitudes, and ~ 0.45 for low latitudes. Values of time-delay at maxima correlation and goodness of fit are given in Tab. 3. Dependencies on solar cycle are similar for both high latitudes, so we modulate the northern parameterization (Eq. 4) for the fitting scheme of the southern parameterization (Eq. 5). The final output is the addition of both sub-daily and 1-10 days parameterizations. The resulting goodness of fits is provided in Tab. 3, and the parameterizations in supporting information files.

Fig. 6 shows the resulting parameterization of *Am* index to represent density disturbances (δ < 10 days) during 2006. The seasonal dependence (Fig. 4d) is clearly seen with lower density disturbances in the SPR ($\rho_S$) around June. On the other hand, density disturbances at the equatorial Region ($\rho_E$) and at the NPR ($\rho_N$) maintain the magnitude of disturbances. Negative density enhancements precursors to geomagnetic storms are not well represented, probably due to a damping response associated with the nitric oxide cooling effects [*Knipp et al.*, 2013]. Note that applying a 10-day running mean

filter, a false negative is introduced in both time-series of proxies and data. This is clearly shown in Fig. 6 with several negative values for the fit of *Am*. However, the residuals have bigger negative values than the *Am* parameterization (Fig. 6). This is due to negative values already present in the time-series (previous to apply the running-mean filter). Previous studies have shown that 37% of CME and 67% of CIR have an abnormal calm state before geomagnetic storms [*Denton et al.*, 2006;
*Borovsky and Steinberg*, 2006]. It has been suggested this effect might be triggered by the Russell-McPherron (RM) effect, through a sector reversal just the upstream of the CIR stream interface.

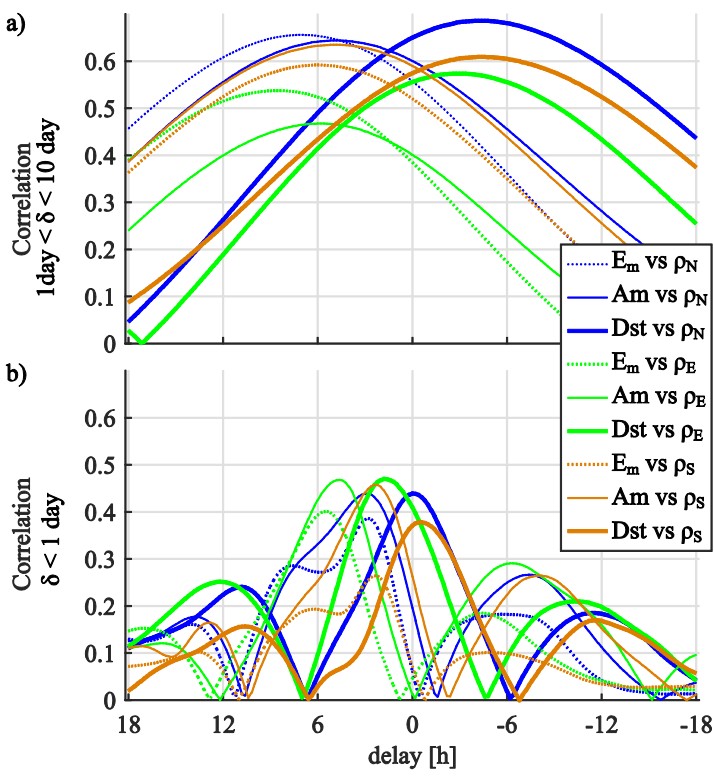

**Figure 5. Delay/correlation between residual density disturbances at 475 km altitude ($\rho_r$) and the parameterizations of disturbances in terms of $E_m$, *Am* and *Dst* indices, for (a) periods between 1 and 10 days and (b) sub-daily periods, for the northern**
**($\rho_N$), equatorial ($\rho_E$), and southern ($\rho_S$) regions.**

**Table 3. Best delay, correlation, and goodness of fit corresponding to Fig. 5.**

| | | | R-square | RMSE [kg/m$^3$] | Correlation | Delay [h] |
|---|---|---|---|---|---|---|
| $1\ \text{day} < \delta < 10\ \text{day}$ | N | Dst | 0.96 | 1.5E-14 | 0.69 | -4.60 |
| | | Am | 0.96 | 1.5E-14 | 0.64 | 4.60 |
| | | E$_m$ | 0.99 | 7.6E-15 | 0.66 | 6.80 |
| | E | Dst | 0.92 | 1.7E-14 | 0.57 | -3.20 |
| | | Am | 0.98 | 7.5E-15 | 0.47 | 5.60 |
| | | E$_m$ | 0.99 | 6.9E-15 | 0.54 | 8.40 |
| | S | Dst | 0.94 | 2.1E-14 | 0.61 | -4.60 |
| | | Am | 0.93 | 2.2E-14 | 0.63 | 4.60 |
| | | E$_m$ | 0.95 | 2.0E-14 | 0.59 | 5.80 |
| $\delta < 1\ \text{day}$ | N | Dst | 0.91 | 1.5E-14 | 0.44 | -0.20 |
| | | Am | 0.92 | 1.4E-14 | 0.44 | 2.80 |
| | | Em | 0.94 | 1.2E-14 | 0.39 | 2.60 |
| | E | Dst | 0.92 | 1.1E-14 | 0.47 | 1.60 |
| | | Am | 0.99 | 3.8E-15 | 0.47 | 4.40 |
| | | E$_m$ | 0.92 | 1.2E-14 | 0.40 | 5.20 |
| | S | Dst | 0.91 | 2.2E-14 | 0.38 | -0.80 |
| | | Am | 0.94 | 1.9E-14 | 0.46 | 2.20 |
| | | E$_m$ | 0.93 | 1.9E-14 | 0.26 | 2.20 |

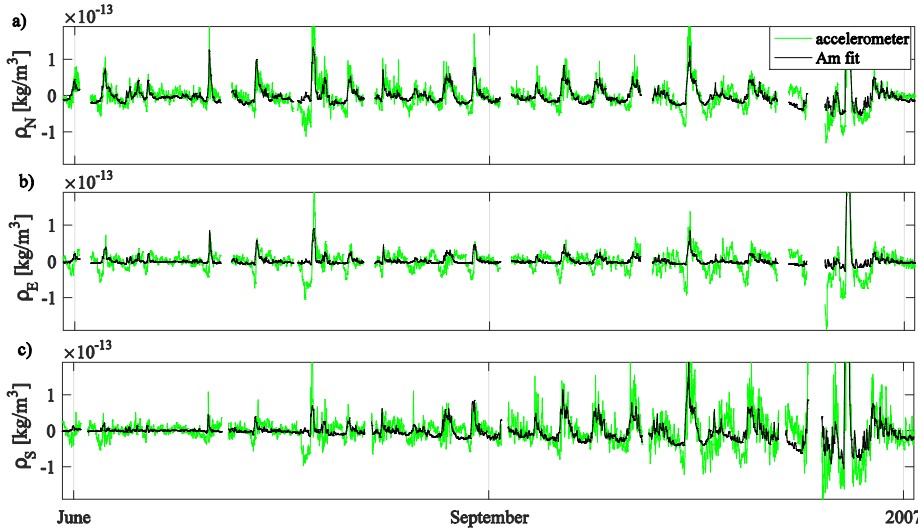

Figure 6. Thermospheric mass density disturbances ($\delta < 10$ days) due to magnetospheric forcing at 475 km altitude, and the parameterization in terms of *Am* index, which is dependent on solar cycle and seasonal variation. From top to bottom, Northern, Equatorial, and Southern profiles are presented. Only June to December on 2006 is presented from the full time-series.

## 3.3. Uncertainty analysis

We further investigate the change in the standard deviation of the residual density disturbances ($\rho_r$) after the removal of parameterized disturbances, to provide an estimate about the uncertainty of the model, through the multiplication of panels (a) to (c) in Fig. 7 with panels (b) to (d) in Fig. 2. In Fig. 7, the reduction in % of the standard deviation (30-day sliding window) is plotted for the northern, the equatorial, and the southern regions. Overall, the results show similar accuracy for all the period investigated, but with some hemispherical difference. The mean value of the reduction of standard deviation is about 30%, and peaks over 50% are seen during all time-series, with slightly lower values at the SPR. The accuracy seems to decrease during low solar activity periods, most probably related to difficulties for fitting low values of disturbances. Taking Fig. 2 as a reference, a reduction in 30% represents a reduction of the standard deviation of about $0.5 \cdot 10^{-13}$ kg/m$^3$ during high solar activity periods, and about $0.06 \cdot 10^{-13}$ kg/m$^3$ during low solar activity periods, being both about the 5% of the background density. The parameterization using the *Dst* index show a larger reduction of residuals at the equator region and mostly during low solar activity, but a larger reduction at high latitude regions are given by the *Am* index under high solar activity conditions. $E_m$ shows the lowest reduction levels during the declining of the solar cycle 23 (2003-2009), but seems to increase during the current solar cycle 24.

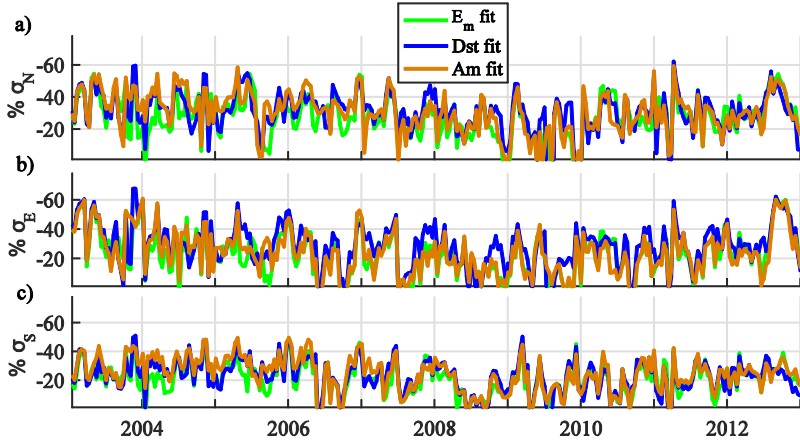

**Figure 7. Reduction in % of the 30-day standard deviation sliding window of residual density disturbances at 475 km altitude ($\rho_r$) when removing the parameterizations in terms of $E_m$, Am and *Dst* indices, and for (a) the northern, (b) the equatorial, and (c) the southern regions. Variations are below 10 days period.**

### 3.4. Contrast of results

Comparing the results given in Tab. 3 with the previous studies, other authors are in good agreement with some particular differences, but no comprehensive analyses were presented concerning differences between SPR and NPR time-lag responses. For instance, *Bruinsma et al.* [2006] showed about 2 h time-delay at high latitudes with respect to solar wind indices and about 4 h for the equatorial regions, and *Rentz and Lühr* [2008] showed about 1 h time-lag with respect to $E_m$. For these authors, we obtain similar results for $E_m$ at sub-daily fluctuations, but a difference of 2 h ahead for longer periods. *Zhou et al.* [2008] showed time lags of about 0-1 h and about 4 h to SYM-H and ∑Q indices, respectively, while our results for *Dst* are also null for sub-daily perturbations, and negative for longer periods (note that *Dst* and SYM-H are similar indices). *Müller et al.* [2009] showed about 3.5 h delay with respect to the *Am* index, and our time-lag for the same index is similar at sub-daily periods, but about 4-7 h for disturbances between 1-10 days period. *Guo et al.* [2010] showed density lag-times of about 3 h for high latitudes and about 4.5 h for low latitudes for IMF-derived indices, and *Liu et al.* [2010, 2011] showed a delay of about 4.5 h for the $E_m$. Our results show about 2 h longer (6-8 h at 1 to 10 days fluctuations). *Zhou et al.* [2013] showed delay times of about 1.5 h, 6 h, and 4.5 h at high, middle and low latitudes, respectively for $E_m$. *Iipponen and Laitinen* [2015] showed 7.5 h and 6 h for the Auroral Electrojet *AE* and the *Ap* indices, respectively, while our results fairly agree with our 6-7 h of time-lag provided by *Am* at 1 to 10 days disturbances. Since the wide range of delay-times provided in the literature does not differentiate the main signal from variations below 24 h, and none of the previous authors has investigated at least a complete solar cycle, we recommend the use of the values provided in Tab. 3.

Finally, we compare our results with three existing upper atmosphere empirical and physical (first-principles) models. We analyze and obtain 10-year profiles of density disturbances from TIEGCM, NRLMSISE-00, and JB2008 in the same way as

we have done with GRACE estimates. TIEGCM 2.0 is computed at 5 min resolution using the 2005 Weimer model [*Weimer*, 2005] using IMF indices to drive high-latitude electric fields. Then, we estimate model densities at the same positions and times than the GRACE measurements along its orbital path. Finally, we employ solar cycle, annual and LST dependencies modeled by the PCA of GRACE and the same filtering techniques detailed in the previous sections. Fig. 8 shows the same plots as Fig. 4 for GRACE results, but only for the variations between 1 and 10 days. The Fourier fits in terms of *doy* and $E_m$ from Fig. 4, and the three models (TIEGCM, NRLMSISE-00, and JB2008) are plotted along with the GRACE results for comparisons. All three models overestimate the disturbance variability at the NPR during low solar activity (2007-2009). During high solar activity (2003-2006 and 2010-2013), the variations seem to fairly agree in all the cases. Variations at the equator are in better agreement with the models, while JB2008 slightly overestimates. These differences are most likely related to a miss-modeled dependence of the 11-year solar cycle variability into the short-term disturbances of magnetospheric-forcing (reefer to results shown in Fig. 2 and 3). This missing contribution shows an imbalance for the magnitude of disturbances between low and high solar-flux periods. Concerning the seasonal variation of the magnitude of disturbances in the SPR, the semi-empirical model JB2008 shows best results, with the best correlation to Fourier fits in terms of *doy* and $E_m$. The assimilation of accelerometer-based densities in the semi-empirical model JB2008 [*Bowman et al.*, 2008] might clearly contribute to better represent the actual thermospheric mass density disturbances due to magnetospheric forcing at the SPR. On the other hand, during low solar activity, TIEGCM and NRLMSISE-00 show bigger magnitude of disturbances during December in the opposite hemisphere (NPR). This feature is not shown from GRACE estimates and less pronounced for JB2008.

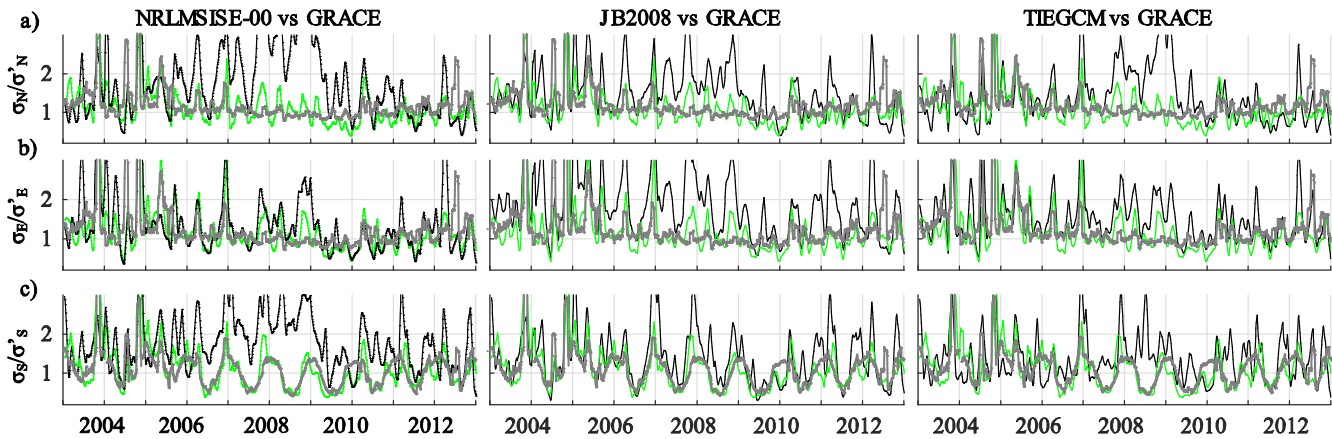

**Figure 8. From top to bottom, the solar-flux normalized 30-day standard deviation sliding window of residual density disturbances at 475 km altitude ($\rho_r$) at (a) the northern, (b) equatorial, and (c) southern regions, for NRLMSISE-00 (left), JB2008 (centered), and TIEGCM (right) models. Data of GRACE (green color) and Fourier fits in terms of *doy* and $E_m$ (gray color) from Fig. 4 are included for comparison. Only variations between 1 and 10 days are represented.**

## 4. Discussions

The hemispherical differential variability of thermospheric mass density disturbances due to the semi-annual variation of geomagnetic activity need to be discussed in relation to the lower disturbances seen during June solstice periods at the SPR (Fig. 4). The equinoctial-axial hypothesis of the semi-annual variation in geomagnetic activity was explained by the semi-annual variation of the effective southward component of the IMF in *Russell and McPherron* [1973]. The RM effect holds that the southward IMF increases when the angle between the Z-axis in geocentric solar magnetospheric (GSM) coordinate system and the Y-axis in geocentric solar equatorial (GSEQ) coordinate system decreases. As mentioned above, a southward IMF will produce a more efficient reconnection and more energy can be introduced into the magnetosphere. This variability can be represented as two maximum around equinoxes, and a minimal around solstices [*Zhao and Zong*, 2012]. We consider very probably this seasonal variation of magnetic range disturbance may transfer high quantities of energy in the MIT system. In fact, *Schaefer et al.* [2016] have shown a similar pattern on the intensity of the Southern Atlantic Anomaly (SAA). The SAA is a large region where the magnetic field is anomalously low and the radiation belt particles reach much lower altitudes than at similar latitudes around the globe. The authors showed that the intensity of the SAA-trapped proton (Van-Allen inner radiation belt) has a minimal around Solstice and maximum during equinox (Fig. 9). In this scheme, our assumptions might induce a tight coupling between the RM variability and the energy transferred to the MIT system, which is seen in these two cases as (1) an increase of energetic particles trapped in the radiation belts, and (2) an increase of energy transferred into the high latitude thermosphere. We, therefore, questioned if a similar pattern could be represented by our residuals in the equatorial and northern regions, and the resulting plot is shown in Fig. 10. In this plot, the residuals are presented to only show the seasonal variability, and a clear similitude to the RM effect [*Zhao and Zong*, 2012] and to the pattern of the SAA intensity (Fig. 9) is shown with minimal values during solstices and maximum values around equinoxes. However, the pattern is more pronounced in the SPR, and a possible explanation can be given from the point of the irregular shape of the Earth's magnetic field.

As mentioned above, the SAA is formed because of the non-coincidence of the southern magnetic dip pole and the Earth's rotating axis. In a similar way, the anomalous low values of the magnetic field in the southern hemisphere during summer might facilitate the energy entrance in the thermosphere, creating relatively higher values during December than during June. In addition, previous studies have found that the extension of SAA decreases during geomagnetic storms, while high-energy protons precipitate from the cusps [*Zou et al.*, 2015]. After a sharp decrease due to a geomagnetic storm, the SAA has shown to recover gradually over several months. Though, since it is questionable in which measure the contribution of the radiation belt can affect the variability of thermospheric mass density disturbances, we address this possible research for future work. In fact, high-energy particles in the Van Allen belts are only a minor source of energy flows into the thermosphere, while the dominant inflows arise from electric fields and auroral particles, such as those linked to the Dungey cycle.

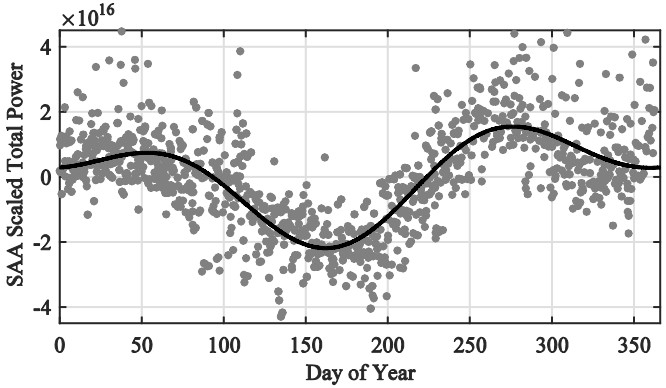

**Figure 9. The SAA intensity changes over the course of a year [*Schaefer et al.*, 2016].**

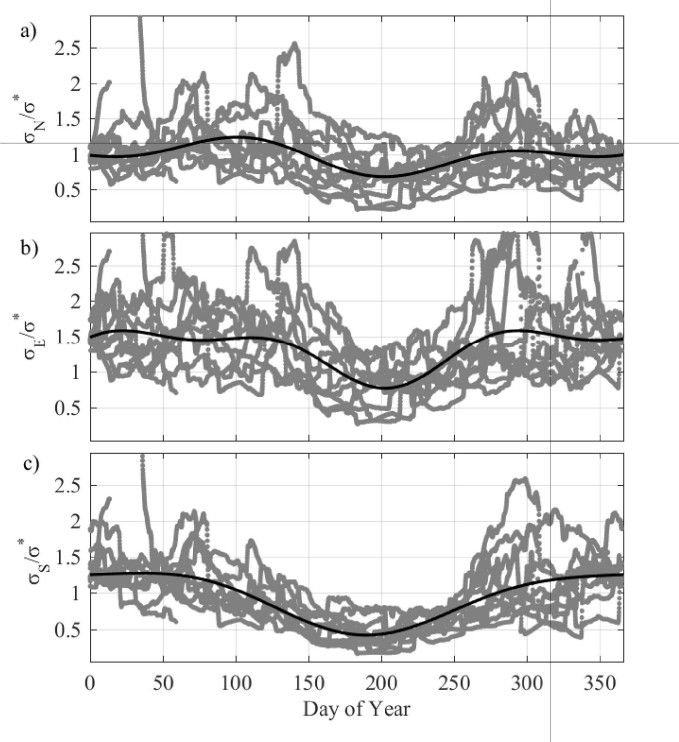

**Figure 10. Normalized residuals from this study showing only the seasonal variation (same as Fig. 4), for (a) the northern, (a) equatorial, and (c) southern regions. Fourier fit in black line.**

Then, under these assumptions and shreds of evidences, the equinox minimal disturbance in terms of the RM effect offers a reasonable explanation for the seasonal variation in the magnitude of mass density disturbances due to magnetospheric forcing (Fig. 10). In addition, the irregular shape of the magnetic field, i.e., offset between the southern dip pole and the rotation axis, might enhance the effects in the SPR, creating the latitudinal asymmetric behavior with enhanced disturbances

during the summer of the southern hemisphere, effect which also is reflected by the SAA. We suspect these enhanced disturbances in the SPR during summer may be caused by an increased energy input through a weaker magnetic field in the noon sector. On the contrary in June solstice, since the northern Earth's magnetic dip pole is located near to the rotation axis ($\sim 3°$), the disturbances may be reduced due to a less compressed Earth's magnetic field. In fact, the evidence of the SAA is a clear example of the effects of the irregular shape of the magnetic field. These results and interpretations are consistent

with the suggestions of an enhanced summer versus winter Joule heating at southern high latitudes of *Bruinsma et al.* [2006]; the very weak anomalies in the SPR during June solstice of *Rentz and Lühr* [2008]; and the 50 % greater dependence of mass density on *Dst* and *Ap* indices on the SPR than that in the NPR shown in *Ercha et al.* [2012].

These results support the potential improvement that can be gained from the use of parametric modeling of the density fluctuations with respect to magnetospheric proxies to improve predictions of thermospheric mass density perturbations, the

455 resulting changes in satellite drag, and other derived physical parameters. Future studies resulting from the removal of mass density disturbances caused by the magnetospheric forcing can be addressed, but not restricted, to investigate additional turbulences, as for example, lower atmospheric waves including tides and planetary waves, recurrent TADs reaching the opposite pole and beyond, or the negative density enhancements during geomagnetic storms.

## 5. Summary

In this study, we have investigated the relationship between indices and mass density disturbances associated with magnetospheric forcing using 10 years (2003-2013) of GRACE observations, after accounting for annual, LST, and solar cycle dependencies through the parameterization of the main PCA modes. In the process, we have removed possible long-term trends in the data by focusing on disturbances on time scales smaller than a 10-day period and divided the analysis into sub-daily disturbances, and those between 1 and 10 days.

The results have shown an unexpected fluctuation of disturbances due to solar cycle variations, and an asymmetric fluctuation with lower values around June solstice at the SPR, hypothetically related to the RM effect and the irregular shape of the Earth's magnetic field. We suspect that in the SPR during summer, when the RM effect is minimal, density enhancements during storm-time periods may be relatively higher than during June, due to increased energy input from a weaker side of the Earth's magnetic field, specifically that which originates the SAA. Notwithstanding, note that the amount

of energy transferred from the Van Allen belts into the thermosphere is only a minor source of energy input, while processes linked to the Dungey cycle may dominate the main variability.

Furthermore, we have detected and parameterized annual and solar cycle dependences included in thermospheric mass density disturbances due to magnetospheric forcing. We employ Pearson's linear correlation coefficients calculated with delay-times ranging from $\pm$ 18 h between estimates and parameterizations at three latitude regions to decipher the best fits.

The parameterization in terms of *Dst* index has shown the best correlation, but without time-delay for prediction. The *Am*

index and $E_m$ have shown great potential as predictors. The $Am$ and $E_m$ indices have provided similar correlation, residuals, and a time-delay of prediction at about 5-8 h. Employing the parameterizations here presented, the reduction of the standard deviation of mass density residual disturbances due to magnetospheric forcing at 475 km altitude reaches a mean value of 30%, and up to 60% of the total residual in several occasions, with respect to residuals from removing only solar cycle, seasonal, and LST dependencies. The parameterizations provided in this manuscript can be re-scaled to required altitude and added to current models, where geomagnetic proxies should be set to $Am=6$ or equivalent. The resulting model is available at http://doi.org/10.5281/zenodo.3234582 [*Calabia and Jin*, 2019].

The main contributions in an easily understood manner are summarized as follows:

- An unexpected dependence on the solar cycle, seasonal variation, and hemispheric asymmetry is found in the magnitude of high frequency (δ < 10 days) thermospheric mass density disturbances due to magnetospheric forcing.

- The seasonal variation produces lower disturbances in June solstice, and the hypothesis of seasonal dependence on the RM effect is presented.

- The hemispheric asymmetry produces higher variability in the SPR, and we suspect a dependence on the irregular shape of the Earth's magnetic field.

- Correlation analysis is conducted using an extensive database (10 years) to provide time-lag values (below 1 h precision) for the currently employed magnetospheric proxies ($Am$, $E_m$, and $Dst$) for thermospheric modeling.

- The high-frequency disturbances (δ < 10 days) have been parameterized in terms of the above dependencies and can be employed to improve current thermospheric models.

These new findings can substantially improve the understanding of the complex MIT system, and help to improve the modeling of thermospheric mass density variations, with the resulting changes in satellite drag.

Comparisons with JB2008, NRLMSISE-00, and TIEGCM models show their incapability to reproduce the seasonal and solar cycle trends of disturbances. Similarities have been found at the equatorial region for the three models, but strong discrepancies during low solar activity for NRLMSISE-00 and TIEGCM showing a model overestimation of disturbance variability. While NRLMSISE-00 overestimates the disturbances during the low solar activity at the SPR, JB2008 shows an impressive agreement with GRACE results, in terms of our hypothesis on the seasonal variation due to the RM effect, and hemispheric asymmetry due to the irregular Earth's magnetic field.

**Author contribution**

A Calabia designed and carried out the experiments, model, and manuscript. SG Jin provided supervision, mentorship, funding support, and revision tasks.

**Acknowledgments**

This work is supported by the National Natural Science Foundation of China-German Science Foundation (NSFC-DFG) Project (Grant No. 41761134092), the Startup Foundation for Introducing Talent of NUIST (Grant No. 2243141801036), and the Talent Start-Up Funding project of NUIST (Grant No. 1411041901010). The authors declare no conflict of interest regarding the publication of this paper. The GRACE data were obtained from the Information System and Data Center

(ISDC) GeoForschungsZentrum (GFZ) website (http://isdc.gfz-potsdam.de/). Mass density estimates and models are provided in the supporting information files [*Calabia and Jin*, 2019].

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
