# Peer review of "Solar cycle, seasonal, and asymmetric dependencies of thermospheric mass density disturbances due to magnetospheric forcing"

_Annales Geophysicae, 2019_

## Referee Comment (RC1) · Anonymous Referee #1 · 4 Jul 2019

**General comments:**

The authors investigate the relations between thermospheric neutral mass densities and geomagnetic indices. The study is based on the evaluation of so-called residual disturbances of the neutral densities obtained from the GRACE mission. Studying the coupling processes in the Earth's atmosphere is a current topic, especially since in-situ measurements of thermospheric neutral densities are available from satellite accelerometer data. The results of this study may have potential to improve the understanding of the thermosphere-magnetosphere coupling. However, the analysis section is not well-described and difficult to read. In large parts, the methodology refers to

the authors' previous publications. Without reading the author's previous papers from 2016, it is hard to get an idea of the applied methods.

Specific Comments:

My main concern about the manuscript is the analysis section. In section 2.1, the authors cite three publications of their own on the processing of the GRACE densities. I recommend to briefly summarize the procedure with a focus on differences compared to other papers. One important point is that the densities have been normalized to an altitude of 475km using densities from the NRLMSISE-00 model. The authors mention in their 2016 publication that this leads to errors of up to 5%. Please discuss this issue.

In section 2.2, the principle modes of neutral densities obtained from 13 years from GRACE accelerometry and POD (from the authors' previous 2016 publication) were used. Why do the PCA modes of 92 %, 3.5 %, 3 %, and 1.3 % listed here differ from the values of 90.3%, 3.5%, 2.9%, 1.2%, listed in Calabia and Jin (2016)? Since the GRACE densities are given along the orbit, it would be interesting to mention the data set on which the PCA has been applied. Did you do a sparse PCA or did you expand the GRACE densities to global grids? Moreover, it is not self-explanatory what the authors mean by "the correlation coefficients between the parameterized time series of PCA modes and the originals". How did you parameterize the time series or is it the principle component itself? What are the originals? Why to the authors add the constant value of Am=6 to which kind of parameterization? How did they choose the value? Additionally, a concise explanation of the "radiation model" would be helpful. Which frequencies did you use, what about semi-annual variations and variations with the solar rotation? Then, the readers should be informed about the reasons for the computation of residual disturbances. What is the benefit from using residual disturbances for the analysis of this paper instead of directly using the GRACE-derived densities? What are the expectations from removing the "radiation model" from the GRACE-derive densities? What about errors in the processing of the GRACE-derived densities? Is there an impact of normalizing the GRACE-derived densities to an altitude of 475km? Are the
density disturbance a temporal or a spatial quantity?

In section 2.3, the residual disturbances are computed for different regions (north, equator, south). What is the difference between density residuals and residual disturbance (both are abbreviated with \rho\_r in section 2.2)? What is the benefit of choosing these three regions? What is the dimension of the density profiles in time and in space (gridded or along the orbit)? Do equations 4 and 5 fit the residual disturbances or their standard deviation? Why does the variable \rho\_r' not occur again in this paper? Since equations 4 and 5 are essential for this paper, I suggest to better motivate and describe this procedure.

Figure 5: The authors' GRACE-derived densities have a resolution of 3 minutes, however, it is possible to compute densities with a resolution of 10seconds. Why do you choose this comparably coarse resolution of 3 minutes? How do you conduct the correlation analysis, i.e., are the indices interpolated to 3 minute intervals? Please discuss the temporal resolution of the indices. Is it reasonable to discuss sub-daily frequencies?

Since many questions arise when reading the analysis section, a lot of effort is required to improve the manuscript. Although the manuscript is well structured, it is difficult to read. I highly recommend to check the language (especially grammar) by a native speaker. Additionally, concise formulations would improve the readability. Typos in citations should not occur (e.g., Lüh  $\rightarrow$  Lühr, Muller  $\rightarrow$  Müller) and abbreviations should be mentioned before using them (e.g., IMF, NPR).

**Technical Corrections:**

Abstract Line 10 Local-Solar-Time  $\rightarrow$  Local Solar Time Line 10 solar-cycle fluctuations  $\rightarrow$  solar cycle variations. Please write "solar cycle" instead of "solar-cycle" also elsewhere. Line 11 ... and investigated in terms of magnetospheric drivers. Please write this into a full sentence. Line 13 weaker variations: This formulation is not clear to me. Maybe you mean low-frequency variations? Please rewrite. Line 14 RussellANGEOD
McPherron (R-M): Abbreviation R-M is not common. Please use RM. Line 19 Dst shows good correlation, while Am and E m are best predictors. It is not clear to me, why Dst shows better correlations than other indices and at the same time it is a worse predictor? Line 18. The reader might not now what Dst, Am and Em stand for. I suggest to rewrite include the meaning of the indices as follows (also for Am and Em): "... in terms of the disturbance storm time index Dst shows ..." Line 18 "Good correlation" is really vague. Adding a number here would be appropriate.

1. Introduction line 26 IMF. Please write out the abbreviation when first mentioning it. Line 27 phenomena. Phenomena is plural. Please change it to phenomenon. Line 37 CME's  $\rightarrow$  CMEs Line 43 equator-ward  $\rightarrow$  towards the equator Line 30-44 Please cite your references! I do not see the point in explaining CIRs and CEMs in detail here, since these phenomena are not relevant for the analysis and conclusions in this study. Line 45 Existing thermospheric modeling  $\rightarrow$  existing thermosphere models Line 46 Low Earth Orbit (LEO) Precise Orbit Determination (POD) → precise orbit determination (POD) of low Earth orbit (LEO) satellites. Lines 46-51. Potential readers might wonder why thermospheric density variations are of great importance for POD. Please add an explanation. Line 52 NRLMSISE00  $\rightarrow$  NRLMSISE-00 Line 51-54. The authors mention three different models that can provide the thermospheric mass density. Why do you think that these models are the most representative ones? For the sake of completeness, please also mention the DTM-2013 by Sean Bruinsma, which is also a representative model. Bruinsma, S. (2015). The DTM-2013 thermosphere model. Journal of Space Weather and Space Climate, 5, A1. Line 54-56. As far as I understand, the mentioned functions are used to model the vertical profile. Please mention this explicitly. Line 60 Liu et al. [2005] is not the first paper on mass densities from CHAMP accelerations. To my knowledge, Villain (1980) was an early paper on densities derived from the CACTUS accelerometer. Then, with the CHAMP mission, Bruinsma and Biancale (2003) published -as far as I know- the first CHAMP-derived mass densities Please clarify this in your literature review: (a) Villain, J.P., 1980. Traitement des donnYees brutes de l'accelerometre CACTUS. Etude des perturbations de
moyenne e Y chelle de la densite thermospherique. Ann. Geophys. 36, 41–49. (b) Bruinsma, S., Biancale, R., 2003. Total density retrieval with STAR 2003. On board evaluation of the STAR accelerometer. In: Reigber, Ch., Lühr, H., Schwintzer, P. (Eds.), First CHAMP Mission Results for Gravity, Magnetic and Atmospheric Studies, Springer, Berlin, Heidelberg, New York, pp. 193–200. Line 61 Please reformulate using proper English. Line 66 Lüh  $\rightarrow$  Lühr Line 70 Muller  $\rightarrow$  Müller. I highly recommend to look at typos in all citations. Line 71 The authors cite Müller et al. (2009) using the Am instead of the Ap index. I agree that this is an important finding, but the difference between the indices remains unclear. Line 97 & 100 The expression "for a better understanding" occurs in two subsequent sentences. I suggest replacing one of them. 104 I suggest to start a new paragraph here to simplify the readability. Line 105 POD -based  $\rightarrow$ POD-based

2. Data and analysis methods Line 117 near-circular orbit  $\rightarrow$  nearly circular orbit Line 117-118 "The highly sensitive accelerometers on-board the GRACE satellites were originally designed to measure the Earth's gravity field..." This sentence is misleading because the accelerometers were not designed to directly measure the Earth's gravity field. The instruments measure the non-gravitational accelerations! Line 172 furrier fitting? Most likely the authors mean a Fourier transform here. line 210, Figure 1. Please put the legend next to the plot to make it fully visible to the reader. Line 260, Figure 4. In comparison to the previous figures, this figure uses bold font in legend and labels. Please do not change the layout of the figures in one paper and stick to normal font.

3. Results and analysis Table 2: Why is there a column \sigma^"\_N that does not include any information? Figure 8 (and its discussion starting at line 344): The overestimation of different density models with respect to different in-situ measurements (from GRACE, CHAMP and Swarm) has also been discussed in other studies. Please address previous findings. See for example: (a) Emmert, J. T. (2015). Thermospheric mass density: A review. Advances in Space Research, 56(5), 773-824. (b) Bruinsma,
S. L., Doornbos, E., & Bowman, B. R. (2014). Validation of GOCE densities and evaluation of thermosphere models. Advances in Space Research, 54(4), 576-585. (c) Mehta, P. M., Walker, A. C., Sutton, E. K., & Godinez, H. C. (2017). New density estimates derived using accelerometers on board the CHAMP and GRACE satellites. Space Weather, 15(4), 558-576.

5. Summary and Conclusions Line 430 is minimum  $\rightarrow$  is minimal Line 431 may relatively be higher. This is no proper English, please rewrite.

---

## Referee Comment (RC2) · Anonymous Referee #2 · 19 Jul 2019

Title: Solar-cycle, seasonal, and asymmetric dependencies of thermospheric mass density disturbances due to magnetospheric forcing Author(s): Andres Calabia and Shuanggen Jin MS No.: angeo-2019-78 MS Type: Regular paper Special Issue: Satellite observations for space weather and geo-hazard

———————————————— General comments

The subject of the paper is the investigation of the relationship between solar and magnetospheric indices and thermosphere density disturbances associated with solar/magnetospheric forcing using GRACE observations. I believe that the analysis proposed by the authors is particularly interesting for two reasons: 1) it uses estimates

of the thermospheric mas density derivted from the high sensitivity accelerometers on board the GRACE mission; 2) analyses the extremely interesting period 2003-2013 containing the descending phase of solar activity, after the second maximum of the solar cycle 23, up to the beginning of the ascending phase of cycle 24, including the exceptional and extended minimum of 2009.

Below I list (not in order of importance) a number of comments on the results and discussed issues.

———————————— Specific comments:

L.33 "Different sunspots regions in the solar corona provide different speeds and densities of solar wind, forming an outward spiral with fast-moving and slow-moving streams". Actually, open/close field regions in the corona are crucial for generation of fast/slow wind more than sunspots regions. I suggest to reformulate as, for example, "Different open and closed magnetic flux domains in the solar corona provide different speeds and densities of solar wind, forming an outward spiral with fast-moving and slow-moving streams"

L.43 "…in thermospheric composition, temperature, density, and winds. Please, add bibliography

L.65 "…showing density increases up to 800 %." Please, specify time scale.

L.120 "…are computed using the methods developed in Calabia and Jin [2016], and are provided at 3 min interval sampling in the supporting information files." I agree with the authors that it is possible to refer to another paper the detailed description of the method. However, I suggest to insert some detail to shortly illustrate the methods used to estimate the mass density.

L.124 The merging electric field Em is introduced without any explanation, I suggest to explain why it is introduced in this point of the paper.

L.131 The authors use the PCA technique to remove "external" forcing to thermosphere data. However, other techniques more suitable for non-stationary signals, e.g., wavelets or EMD, have been used by other authors. Please, shortly describe and support the choice of PCA method.

L.150 Please justify the choice of a 10d period. Why not, for example, a half solar rotation period (about 12d).

L.187 Section 3. I strongly suggest a reorganization of the whole section 3. At the moment it seems an unorganized list of figure comments related to analysis of single events (e.g., Fig.1), particular observation periods (e.g., Fig.6) or whole period (e.g., Fig.4). I would suggest structuring it with subsections, by listing the various analyses that have been carried out, the objectives and the results.

L240-L255 This part is confusing (with several references to different thermosphere regions and different figures, tables and equations). It must be rewritten by ordering the analysis and the results. Moreover in L.249 (and Fig.4 and Table 2 captions) is introduced an unknown Furrier fit. I suppose you mean Fourier fit.

L.369 Section 4: Perhaps it can be redistributed in a section 3 (Data analysis and discussion) reorganized into subsections.

L.445-450 better at the end of the section.

————————————————- Technical corrections:

L.133 "The aim of by 'a' PCA technique is to determine..." suggested "The purpose/aim of a PCA technique is to determine..."

L.137 Please, reshape the sentence "The highly values of explained variance for the first modes indicate marked patterns of variability, and the correlations to parameterizations indicate high accuracy in the model." in order to clarify the concept.

L.141/L.144 residuals (r) or residual disturbances (r)? I suggest to use the same definition for the same symbol (r).

L.150 "Density residuals at three latitude regions are..." the authors can specify here the chosen latitudes.

L.158 "...to fit the direct signal of disturbances." do you mean "residual disturbances (r)"?

L.248 ...standard deviations ($\sigma$"). => ...standard deviations ($\sigma$).

L.249 (and Fig.4 and Table 2 captions) is introduced an unknown Furrier fit. I suppose you mean Fourier fit.

L.390 magnetic dip pole and the Earth's => magnetic dipole and the Earth's

L.406 dip pole => dipole

---

## Author Response (AR1)

**General comments:**

The authors investigate the relations between thermospheric neutral mass densities and geomagnetic indices. The study is based on the evaluation of so-called residual disturbances of the neutral densities obtained from the GRACE mission. Studying the coupling processes in the Earth's atmosphere is a current topic, especially since in-situ measurements of thermospheric neutral densities are available from satellite accelerometer data. The results of this study may have potential to improve the understanding of the thermosphere-magnetosphere coupling. However, the analysis section is not well-described and difficult to read. In large parts, the methodology refers to

the authors' previous publications. Without reading the author's previous papers from 2016, it is hard to get an idea of the applied methods.

Specific Comments:

My main concern about the manuscript is the analysis section. In section 2.1, the authors cite three publications of their own on the processing of the GRACE densities. I recommend to briefly summarize the procedure with a focus on differences compared to other papers. One important point is that the densities have been normalized to an altitude of 475km using densities from the NRLMSISE-00 model. The authors mention in their 2016 publication that this leads to errors of up to 5%. Please discuss this issue.

In section 2.2, the principle modes of neutral densities obtained from 13 years from GRACE accelerometry and POD (from the authors' previous 2016 publication) were used. Why do the PCA modes of 92 %, 3.5 %, 3 %, and 1.3 % listed here differ from the values of 90.3%, 3.5%, 2.9%, 1.2%, listed in Calabia and Jin (2016)? Since the GRACE densities are given along the orbit, it would be interesting to mention the data set on which the PCA has been applied. Did you do a sparse PCA or did you expand the GRACE densities to global grids? Moreover, it is not self-explanatory what the authors mean by "the correlation coefficients between the parameterized time series of PCA modes and the originals". How did you parameterize the time series or is it the principle component itself? What are the originals? Why to the authors add the constant value of Am=6 to which kind of parameterization? How did they choose the value? Additionally, a concise explanation of the "radiation model" would be helpful. Which frequencies did you use, what about semi-annual variations and variations with the solar rotation? Then, the readers should be informed about the reasons for the computation of residual disturbances. What is the benefit from using residual disturbances for the analysis of this paper instead of directly using the GRACE-derived densities? What are the expectations from removing the "radiation model" from the GRACE-derive densities? What about errors in the processing of the GRACE-derived densities? Is there an impact of normalizing the GRACE-derived densities to an altitude of 475km? Are the
density disturbance a temporal or a spatial quantity?

In section 2.3, the residual disturbances are computed for different regions (north, equator, south). What is the difference between density residuals and residual disturbance (both are abbreviated with \rho\_r in section 2.2)? What is the benefit of choosing these three regions? What is the dimension of the density profiles in time and in space (gridded or along the orbit)? Do equations 4 and 5 fit the residual disturbances or their standard deviation? Why does the variable \rho\_r' not occur again in this paper? Since equations 4 and 5 are essential for this paper, I suggest to better motivate and describe this procedure.

Figure 5: The authors' GRACE-derived densities have a resolution of 3 minutes, however, it is possible to compute densities with a resolution of 10seconds. Why do you choose this comparably coarse resolution of 3 minutes? How do you conduct the correlation analysis, i.e., are the indices interpolated to 3 minute intervals? Please discuss the temporal resolution of the indices. Is it reasonable to discuss sub-daily frequencies?

Since many questions arise when reading the analysis section, a lot of effort is required to improve the manuscript. Although the manuscript is well structured, it is difficult to read. I highly recommend to check the language (especially grammar) by a native speaker. Additionally, concise formulations would improve the readability. Typos in citations should not occur (e.g., Lüh  $\rightarrow$  Lühr, Muller  $\rightarrow$  Müller) and abbreviations should be mentioned before using them (e.g., IMF, NPR).

**Technical Corrections:**

Abstract Line 10 Local-Solar-Time  $\rightarrow$  Local Solar Time Line 10 solar-cycle fluctuations  $\rightarrow$  solar cycle variations. Please write "solar cycle" instead of "solar-cycle" also elsewhere. Line 11 ... and investigated in terms of magnetospheric drivers. Please write this into a full sentence. Line 13 weaker variations: This formulation is not clear to me. Maybe you mean low-frequency variations? Please rewrite. Line 14 RussellANGEOD
McPherron (R-M): Abbreviation R-M is not common. Please use RM. Line 19 Dst shows good correlation, while Am and E m are best predictors. It is not clear to me, why Dst shows better correlations than other indices and at the same time it is a worse predictor? Line 18. The reader might not now what Dst, Am and Em stand for. I suggest to rewrite include the meaning of the indices as follows (also for Am and Em): "... in terms of the disturbance storm time index Dst shows ..." Line 18 "Good correlation" is really vague. Adding a number here would be appropriate.

1. Introduction line 26 IMF. Please write out the abbreviation when first mentioning it. Line 27 phenomena. Phenomena is plural. Please change it to phenomenon. Line 37 CME's  $\rightarrow$  CMEs Line 43 equator-ward  $\rightarrow$  towards the equator Line 30-44 Please cite your references! I do not see the point in explaining CIRs and CEMs in detail here, since these phenomena are not relevant for the analysis and conclusions in this study. Line 45 Existing thermospheric modeling  $\rightarrow$  existing thermosphere models Line 46 Low Earth Orbit (LEO) Precise Orbit Determination (POD) → precise orbit determination (POD) of low Earth orbit (LEO) satellites. Lines 46-51. Potential readers might wonder why thermospheric density variations are of great importance for POD. Please add an explanation. Line 52 NRLMSISE00  $\rightarrow$  NRLMSISE-00 Line 51-54. The authors mention three different models that can provide the thermospheric mass density. Why do you think that these models are the most representative ones? For the sake of completeness, please also mention the DTM-2013 by Sean Bruinsma, which is also a representative model. Bruinsma, S. (2015). The DTM-2013 thermosphere model. Journal of Space Weather and Space Climate, 5, A1. Line 54-56. As far as I understand, the mentioned functions are used to model the vertical profile. Please mention this explicitly. Line 60 Liu et al. [2005] is not the first paper on mass densities from CHAMP accelerations. To my knowledge, Villain (1980) was an early paper on densities derived from the CACTUS accelerometer. Then, with the CHAMP mission, Bruinsma and Biancale (2003) published -as far as I know- the first CHAMP-derived mass densities Please clarify this in your literature review: (a) Villain, J.P., 1980. Traitement des donnYees brutes de l'accelerometre CACTUS. Etude des perturbations de
moyenne e Y chelle de la densite thermospherique. Ann. Geophys. 36, 41–49. (b) Bruinsma, S., Biancale, R., 2003. Total density retrieval with STAR 2003. On board evaluation of the STAR accelerometer. In: Reigber, Ch., Lühr, H., Schwintzer, P. (Eds.), First CHAMP Mission Results for Gravity, Magnetic and Atmospheric Studies, Springer, Berlin, Heidelberg, New York, pp. 193–200. Line 61 Please reformulate using proper English. Line 66 Lüh  $\rightarrow$  Lühr Line 70 Muller  $\rightarrow$  Müller. I highly recommend to look at typos in all citations. Line 71 The authors cite Müller et al. (2009) using the Am instead of the Ap index. I agree that this is an important finding, but the difference between the indices remains unclear. Line 97 & 100 The expression "for a better understanding" occurs in two subsequent sentences. I suggest replacing one of them. 104 I suggest to start a new paragraph here to simplify the readability. Line 105 POD -based  $\rightarrow$ POD-based

2. Data and analysis methods Line 117 near-circular orbit  $\rightarrow$  nearly circular orbit Line 117-118 "The highly sensitive accelerometers on-board the GRACE satellites were originally designed to measure the Earth's gravity field..." This sentence is misleading because the accelerometers were not designed to directly measure the Earth's gravity field. The instruments measure the non-gravitational accelerations! Line 172 furrier fitting? Most likely the authors mean a Fourier transform here. line 210, Figure 1. Please put the legend next to the plot to make it fully visible to the reader. Line 260, Figure 4. In comparison to the previous figures, this figure uses bold font in legend and labels. Please do not change the layout of the figures in one paper and stick to normal font.

3. Results and analysis Table 2: Why is there a column \sigma^"\_N that does not include any information? Figure 8 (and its discussion starting at line 344): The overestimation of different density models with respect to different in-situ measurements (from GRACE, CHAMP and Swarm) has also been discussed in other studies. Please address previous findings. See for example: (a) Emmert, J. T. (2015). Thermospheric mass density: A review. Advances in Space Research, 56(5), 773-824. (b) Bruinsma,
S. L., Doornbos, E., & Bowman, B. R. (2014). Validation of GOCE densities and evaluation of thermosphere models. Advances in Space Research, 54(4), 576-585. (c) Mehta, P. M., Walker, A. C., Sutton, E. K., & Godinez, H. C. (2017). New density estimates derived using accelerometers on board the CHAMP and GRACE satellites. Space Weather, 15(4), 558-576.

5. Summary and Conclusions Line 430 is minimum  $\rightarrow$  is minimal Line 431 may relatively be higher. This is no proper English, please rewrite.
Ann. Geophys. Discuss., https://doi.org/10.5194/angeo-2019-78-AC3, 2019 © Author(s) 2019. This work is distributed under the Creative Commons Attribution 4.0 License.

ANGEOD
Thank you very much to Editor and Reviewers for this review where constructive comments and valuable suggestions have greatly improved the quality and content of the manuscript. In the followings, comments and suggestions of the two Reviewers, as well as our replies to them, are given. We hope that the revised version of the manuscript together with our replies, cover the Reviewers' comments appropriately and, of course, your further concerns and advice are appreciated.

Best Regards,

Andres Calabia and Shuanggen Jin

Answers to Reviewer 1:

1. In section 2-1, we have briefly summarized previous work, methodology, and included relevant references.

2. We have included a reference on the issue of density normalization.

3. In this work, more data and a revised analysis have been performed with respect to the work done in Calabia and Jin [2016]. For instance, we have included POD-based estimates to fill the data gaps of accelerometer measurements [Calabia and Jin, 2017], and manually excluded outliers caused by, e.g., geomagnetic storms and artifacts in the data processing. These improvements have provided a better representation of the variability with respectCalabia and Jin (2016) results.

4. Since GRACE densities are given along the orbit, we have derived global grids by interpolating the initial values. Calabia and Jin (2016) provide a set of grids in the supporting information.

5. We have separately parameterized the PCA time series in terms of solar-flux, annual, and Local Solar Time dependencies (also include other minor dependencies, e.g., daily Am, P, and K waves). Then, we employ the correlation between each parameterized component and the initial time-series.

6. We employ a constant value of Am=6 to set an arbitrary constant quiet-time value to the parameterization, so we can investigate in detail the magnetospheric contribution in the residuals. The parameterization scheme is based on daily grids, so sub-daily variations are therefore averaged to a daily contribution.

7. The "radiation model" is basically the parameterization scheme (Calabia and Jin,2016) excluding magnetospheric contribution. The frequencies are extracted by
frequency analysis (periodogram). Annual, semi-annual, etc variations are included in Fourier series.

8. The residual disturbances for the analysis of this paper are obtained by removing the "radiation model" from the GRACE-derive densities. In this way, contributions due to solar flux radiation, annual, and LST variations are removed from the time-series.

9. A brief introduction is given for the errors in the processing of the GRACE-derived densities. Further reading can be found in the references.

10. In this work, density residual disturbances have temporal and spatial dimensions. In the time dimension, residual disturbances follow the variations of magnetospheric activity. In the spatial dimension, residual disturbances originate at high latitude regions and propagate towards the equator.

11. In section 2.3, the residual disturbances are computed for different regions (north, equator, south). The benefit of choosing these three regions is the possibility to study the propagation time of density disturbances from high-latitude to equatorial regions.

12. In this manuscript, density residuals and residual disturbance (both are abbreviated with rho\_r in section 2.2) are the same things.

13. The dimension in time and space of the density profiles corresponds to the gridded dimension.

14. Equations 4 and 5 fit the direct residual disturbances. The variable rho\_r' is the general form of the three profiles (north, equator, south). Equations 4 and 5 are the polynomial form of the empirical fitting.

15. Figure 5: GRACE-derived densities along the orbit (at 1 sec. interval, these are the initial estimates) have been re-sampled at a resolution of 3 minutes to reduce the size of the file given in supporting information. The gridded data are derived from 1-second sampling estimates. Two sets of grids are derived, one for density estimates and other for measurement-times(at each pixel of the grids). The profiles are derived

ANGEOD
from these grids. For the correlation analysis, magnetospheric indices are interpolated to the same times given by each corresponding pixel. It could be reasonable to discuss/study sub-daily frequencies but should consider possible additional dependencies (e.g., orbit precession, geographical location).

16. The analysis section has been revised to improve reader understanding.

17. Typos incitations and abbreviations have been revised.

Corrections have been revised according to your suggestions and comments. Here some details and additional clarifications:

Line 13 "weaker variations"-> "smaller amplitudes".

Line 19 "Dstshows good correlation, while Am and Em are best predictors."Dst is a worse predictor because does not provide time delay to predict in real-time.

Line 18 "Good correlation" -> "best correlation". We prefer no adding a number since the values compared to previous authors might lead to misunderstanding of its actual assessment. Explicitly, "none of the previous authors have investigated a sufficiently large and continuous time-series of observations, at least to complete a solar cycle, while their statistical analyses were focused only on collections of large storms". In this scheme, the correlation index for a complete time-series provides lower correlation values than that performed by a collection of events.

Line 30-44 CIRs and CEMs are briefly introduced here to explain the origin of short-term density disturbances.

Lines 46-51. Added explanation: "Aerodynamic-drag associated with neutral-density fluctuations resulting from upper atmospheric expansion/contraction in response to variable solar and geomagnetic activity, increases drag and decelerates Low Earth Orbits, the dwindling lifespan of space-assets, and making tracking difficult."

In Figure 1, we consider more convenient to put the legend inside the plot to make the

ANGEOD
graphic bigger.

In Figure 4, we do not use bold font in legend and labels. The fonts might appear ticker due to software issues.

In Table 2, we have removed the column for the north profile. Note that north and equator profiles have not been parameterized for annual variation.

Figure 8 (and its discussion starting at line 344): Note that this work does not study the "absolute" over-estimation of different density models with respect to different in-situ measurements as done by other studies. Instead, it analyzes and obtains 10-year "profiles of density disturbances" from models in the same way as we have done with GRACE estimates.

Please also note the supplement to this comment: https://www.ann-geophys-discuss.net/angeo-2019-78/angeo-2019-78-AC3supplement.pdf

ANGEOD
Ann. Geophys. Discuss., https://doi.org/10.5194/angeo-2019-78-RC2, 2019 © Author(s) 2019. This work is distributed under the Creative Commons Attribution 4.0 License.

ANGEOD
Title: Solar-cycle, seasonal, and asymmetric dependencies of thermospheric mass density disturbances due to magnetospheric forcing Author(s): Andres Calabia and Shuanggen Jin MS No.: angeo-2019-78 MS Type: Regular paper Special Issue: Satel-lite observations for space weather and geo-hazard

General comments

The subject of the paper is the investigation of the relationship between solar and magnetospheric indices and thermosphere density disturbances associated with solar/magnetospheric forcing using GRACE observations. I believe that the analysis proposed by the authors is particularly interesting for two reasons: 1) it uses estimates

of the thermospheric mas density derivted from the high sensitivity accelerometers on board the GRACE mission; 2) analyses the extremely interesting period 2003-2013 containing the descending phase of solar activity, after the second maximum of the solar cycle 23, up to the beginning of the ascending phase of cycle 24, including the exceptional and extended minimum of 2009.

Below I list (not in order of importance) a number of comments on the results and discussed issues.

—— Specific comments:

L.33 "Different sunspots regions in the solar corona provide different speeds and densities of solar wind, forming an outward spiral with fast-moving and slow-moving streams". Actually, open/close field regions in the corona are crucial for generation of fast/slow wind more than sunspots regions. I suggest to reformulate as, for example, "Different open and closed magnetic flux domains in the solar corona provide different speeds and densities of solar wind, forming an outward spiral with fast-moving and slow-moving streams".

L.43 "...in thermospheric composition, temperature, density, and winds. Please, add bibliography

L.65 "... showing density increases up to 800 %." Please, specify time scale.

L.120 "...are computed using the methods developed in Calabia and Jin [2016], and are provided at 3 min interval sampling in the supporting information files." I agree with the authors that it is possible to refer to another paper the detailed description of the method. However, I suggest to insert some detail to shortly illustrate the methods used to estimate the mass density.

L.124 The merging electric field Em is introduced without any explanation, I suggest to explain why it is introduced in this point of the paper.

L.131 The authors use the PCA technique to remove "external" forcing to thermo-
sphere data. However, other techniques more suitable for non-stationary signals, e.g., wavelets or EMD, have been used by other authors. Please, shortly describe and support the choice of PCA method.

L.150 Please justify the choice of a 10d period. Why not, for example, a half solar rotation period (about 12d).

L.187 Section 3. I strongly suggest a reorganization of the whole section 3. At the moment it seems an unorganized list of figure comments related to analysis of single events (e.g., Fig.1), particular observation periods (e.g., Fig.6) or whole period (e.g., Fig.4). I would suggest structuring it with subsections, by listing the various analyses that have been carried out, the objectives and the results.

L240-L255 This part is confusing (with several references to different thermosphere regions and different figures, tables and equations). It must be rewritten by ordering the analysis and the results. Moreover in L.249 (and Fig.4 and Table 2 captions) is introduced an unknown Furrier fit. I suppose you mean Fourier fit.

L.369 Section 4: Perhaps it can be redistributed in a section 3 (Data analysis and discussion) reorganized into subsections.

L.445-450 better at the end of the section.

-- Technical corrections:

L.133 "The aim of by 'a' PCA technique is to determine. . . " suggested "The purpose/aim of a PCA technique is to determine. . . "

L.137 Please, reshape the sentence "The highly values of explained variance for the first modes indicate marked patterns of variability, and the correlations to parameterizations indicate high accuracy in the model." in order to clarify the concept.

L.141/L.144 residuals (r) or residual disturbances (r)? I suggest to use the same definition for the same symbol (r).

ANGEOD
L.150 "Density residuals at three latitude regions are..." the authors can specify here the chosen latitudes.

L.158 "...to fit the direct signal of disturbances." do you mean "residual disturbances (r)"?

```
L.248 ... standard deviations (\sigma"). => ... standard deviations (\sigma).
```

L.249 (and Fig.4 and Table 2 captions) is introduced an unknown Furrier fit. I suppose you mean Fourier fit.

L.390 magnetic dip pole and the Earth's => magnetic dipole and the Earth's

L.406 dip pole => dipole
Ann. Geophys. Discuss., https://doi.org/10.5194/angeo-2019-78-AC2, 2019 © Author(s) 2019. This work is distributed under the Creative Commons Attribution 4.0 License.

ANGEOD
Thank you very much to Editor and Reviewers for this review where constructive comments and valuable suggestions have greatly improved the quality and content of the manuscript. In the followings, comments and suggestions of the two Reviewers, as well as our replies to them, are given. We hope that the revised version of the manuscript together with our replies, cover the Reviewers' comments appropriately and, of course, your further concerns and advice are appreciated.

Best Regards,

Andres Calabia and Shuanggen Jin

Answers to reviewer 2:

L.33 reformulated as suggested.

L.43 ". . .in thermospheric composition, temperature, density, and winds. " bibliography added.

L.65 ". . . showing density increases up to 800 % in a few hours." time scale specified.

L.120 The methods used to estimate the mass density has been briefly introduced and more references have been included.

L.124 The merging electric field Em has been introduced.

L.131 Note that the PCA components are parameterized in terms of annual, LST, and solar flux variations. We remove this "well-known" forcing to thermosphere data, and the residuals are investigated looking for new dependencies. Other techniques such as wavelets could have also been used, but we preferred to employ this two-step method (PCA fit and fit of residuals) for more robust modeling.

L.150 We arbitrarily decided to employ the 10d period. A half solar rotation period (about 12d) would have provided similar results.

L.187 We have structured Section 3 with subsections, and listed the various analyses at the beginning, including the objectives and the results.

L.249 Furrier typo has been corrected.

L.445-450 has been relocated at the end of the section as suggested.

Answers to technical corrections:

L.133 "Change done as suggested ("The purpose/aim of a PCA technique is to deter-
mine. . ." ) L.137 has been reshaped to clarify the concept.

L.141/L.144 residual disturbances used for the symbol (r).

L.150 chosen latitudes specified.

L.158 changed to "residual disturbances (r)".

L.249 Furrier typo is corrected all along the paper.

L.390 and L.406 "dip pole" is employed in these cases instead "dipole".

Please also note the supplement to this comment: https://www.ann-geophys-discuss.net/angeo-2019-78/angeo-2019-78-AC2supplement.pdf